# Analysis of the Functionally Step-Variable Graded Plate Under In-Plane Compression

**DOI:** 10.3390/ma12244090

**Published:** 2019-12-07

**Authors:** Leszek Czechowski, Zbigniew Kołakowski

**Affiliations:** Department of Strength of Materials, Lodz University of Technology, 90-924 Lodz, Poland; zbigniew.kolakowski@p.lodz.pl

**Keywords:** finite element method, functionally graded materials, ceramics, buckling, asymptotic Koiter’s theory

## Abstract

A study of the pre- and post-buckling state of square plates built from functionally graded materials (FGMs) and pure ceramics is presented. In contrast to the theoretical approach, the structure under consideration contains a finite number of layers with a step-variable change in mechanical properties across the thickness. An influence of ceramics content on a wall and a number of finite layers of the step-variable FGM on the buckling and post-critical state was scrutinized. The problem was solved using the finite element method and the asymptotic nonlinear Koiter’s theory. The investigations were conducted for several boundary conditions and material distributions to assess the behavior of the plate and to compare critical forces and post-critical equilibrium paths.

## 1. Introduction

Functionally graded materials (FGMs) are treated as modern materials which due to their varying properties for thickness or other dimensions are assigned to work in ultrahigh hard environments under high and other temperature fields. Gradual changes in the volume fraction of constituents and the non-homogenous structure provide theoretically continuous graded macroscopic properties (hardness, wear resistance, thermal conductivity, specific heat, mass density, etc.). At present, well-known techniques for the manufacturing of FGMs have been strongly developed, however taking into account different properties or different combinations of the components used, none of them can be found to be excellent and devoid of defects [1]. While taking a look at methods of FGM fabrication, one can distinguish: gas-based methods (chemical vapor deposition, thermal spray, and the surface reaction process), liquid phase processes (chemical solution deposition, laser deposition, and electro-chemical gradation) or solid-phase processes (spark plasma sintering, powder metallurgy) [1]. It should be mentioned that actual FGMs differ from one another in contrast to the theoretical descriptions of these materials. Thus, it is impossible to obtain a perfect structure with fluently varying properties. Indeed, moderately adequate structures of FGMs with appropriate constituents can be attained using some techniques, but many methods enable producing mere multilayer (several-layer) moderately thick structures of FGMs (using spark plasma sintering, powder metallurgy, etc.). The basic concept of FGMs was presented by Niino and the other studies referring to FGMs can be found in references [2,3,4,5]. Further research on FGMs showed that many works were devoted to an analysis of manufacturing processing and its influence on various properties [6,7,8,9]. On the other hand, in many papers based on original assumptions, perfect structures were often analyzed (with theoretical and assumed material distributions as well as assumed material properties). Those works concerned theoretical studies on the stability of thin-walled structures under mechanical or/and thermal load (plates, boxes, etc.) or investigations of the frequency of natural vibrations. Such results of buckling and post-buckling of the FGM plate under mechanical loads were included in references [10,11,12,13,14,15,16,17] or were developed for plate structures in reference [18,19,20]. Analyses of natural vibrations of FGM structures were reported in references [21,22,23]. Other works were focused on the investigation of the structure behavior under static or dynamic thermal loads [24,25,26,27]. Kumar et al. in reference [28] analyzed the buckling and post-buckling state of an FGM plate using two new higher-order transverse shear deformation theories (NHSDTs). Trabelsi et al. in reference [29] studied the response of FGM shell structures (plates and cylindrical shells separately) due to thermal load with the first-order shear deformation theory (FOSDT). In reference [30], a study of the thick simply supported FGM plate under bending was presented within the displacement potential function (DPF). Nguyen et al. in their papers [31,32] proposed analytical modeling of thin-walled open-section beams made of functionally graded materials on the basis of Vlasov’s assumptions. Moreover, the same authors also investigated functionally graded open-section beams with different types of material distributions by means of a two-node beam element with 14 degrees of freedom in reference [33]. Xu et al. [34] analyzed elastoplastic buckling behaviors of rectangular plates built of functionally graded materials with a homogenization method of the Tamura-Tomota-Ozawa model. Besides studies on strictly functionally graded materials, recently more and more analyses devoted to printed functional materials have been developed which can be found in references [35,36,37,38], and elsewhere.

As assessed in the literature, almost all of these papers were based on theoretical or mathematical mechanical properties of perfect FGM structures. Because it seems impossible to achieve an idealized material distribution (on the basis of the literature survey), the authors of the present paper conducted investigations and showed the results for a plate with step-variable gradation of mechanical properties and with a finite number of layers across the wall. Moreover, a combination of the FGM with additional components (in the presented case, additional thickness of ceramics was studied) was taken into consideration. The non-linear problem of stability was solved with the semi-analytical method (SAM) based on non-linear Koiter’s theory [27] and with the finite element method (FEM) code ANSYS^®^ [39]. The full Green’s strain tensor, the second Piola-Kirchhoff’s stress tensor, and the transition matrix using Godunov’s orthogonalization were used in the description of the problem. In reference [17], on the basis of Koiter’s theory, FGM plates have non-symmetric stable post-buckling equilibrium paths. This feature explains differences in the plate response dependence on the imperfection sign. An FGM plate has a non-trivial coupling matrix B and the coupling between extensional and bending deformations exists as is in the case of non-symmetric laminated plates. On the basis of both of the above-mentioned methods, the present paper reveals the results of influence on the critical and post-critical state of plates made of an alumina-FGM with a finite number of layers (a total of five to 15 layers were assumed). Furthermore, several boundary conditions of the edge support and different contents of alumina in relation to the FGM were taken into account. Considered variants of material distributions were assumed to reflect real FGM.

## 2. Problem Description

The object of investigation was a square plate (a = b = 1 m) subjected to the mechanical compression load, as shown in Figure 1a.

The length and the total thickness of the plate was equal to 200 mm and 2 mm ($t_{t}$), respectively.

The thickness of pure ceramics $t_{c}$ ranges from 0.2 mm to 1 mm but the thickness of FGM $t_{FGM}$ is equal from 1.0 mm to 1.8 mm (the total thickness of wall is a sum $t_{t}=t_{FGM}+t_{c}$). The description of considered variants were posted in Table 1. The total thickness of a plate comprises the FGM (Al-Al_2_0_3_) and the pure ceramics Al_2_0_3_ (See Figure 1b).

To solve the problem, two methods were employed: an analytical-numerical method based on the asymptotic Koiter’s approach [15,16,17] and a numerical one based on the finite element method [39]. To verify an influence of the material distribution (see Table 1) for Var_1 from 5 up to 15 (5, 7, 11, 15), layers of the FGM with a composition shown in Figure 2 were considered. In the case of Var_2-Var_5, 11 layers of the FGM were assumed. Basic material properties for the considered constituents have been listed in Table 2, but in the case of a composition of different materials in a dependence on their content, the mixture law was applied.

### 2.1. FE Model

Numerical simulations based on the finite element method were conducted with the ANSYS 18.2^®^ software [39]. To generate an adequate numerical model, an 8-node 281 shell element was assumed. The plate was divided into 10 k finite elements (100 elements along the edge—Figure 3). A nonlinear analysis for large deflections was performed on the basis of Green-Lagrangian equations. During computations, nonlinear calculations were conducted in accordance with the Newton-Raphson algorithm. The critical- and post-critical state of the plate with different boundary conditions was analyzed (See Table 3). The initial deflection of the plate in all cases was assumed to be 0.01t_t_, which referred to the first buckling mode.

### 2.2. Koiter’s Asymptotic Approach

The equilibrium equations for FGM plate structures can be written as [15,16,17,18]:(1)$$\left(1-\frac{\sigma }{\sigma_{r}}\right)\zeta_{r}+a_{pqr}\zeta_{p}\zeta_{q}+b_{rrrr}\zeta_{r}^{3}-\frac{\sigma }{\sigma_{r}}\zeta_{r}^{*}+\ldots =0\;\mathrm{for}\;r=1,\ldots J$$
where: $\sigma_{r}$ is the critical stress of the *r-*th buckling mode, $\zeta_{r}$ is the dimensionless amplitude of the *r*-th buckling mode, $\zeta_{r}^{*}$ is the dimensionless amplitude of the initial imperfections related to the *r-*th buckling mode, σ is the compressive stress, and $a_{pqr}$ and $b_{rrrr}$ are the coefficients, respectively. The range of indices are p, q, r is from 1 to J, where J is the number of interacting modes. The summation was determined based on the repeated indices. For the case of the uncoupled buckling mode (i.e., for the one-mode approach), J=1 has to be satisfied. The first order coefficients (i.e., $a_{pqr}$) were found with the analytical-numerical method based on Koiter’s theory [15-18,27] etc. The second order coefficients (i.e., $b_{rrrr}$) were calculated with the semi-analytical method (SAM) [16,17]. When using that method, it was necessary to determine approximate values of $b_{rrrr}$ on the basis of the linear buckling problem. The co-author of SAM is also the co-author of this article. In the present work, a SAM one-modal approach has been applied (J=1 in Equation (1)). This means one degree of freedom in nonlinear analysis was taken into account. The imperfection of the plate was assumed as in the numerical model: $\zeta_{1}^{*}=w_{0}/t_{t}=1.0$. The finite element method allows us to simulate a complex numerical model but boundary conditions and element types also play a major role. In comparison to the SAM included in the present paper, a single calculation based on FEM requires between one hour and a couple of days to compute. Using the semi-analytical method, one can obtain the results within 5–15 min, which can provide a major benefit. Moreover, differences in results between two methods amount to a few percent (referring to critical forces) or to several dozen percent (compared to the post-buckling paths), at most.

## 3. Results and Discussion

### 3.1. Buckling Forces

In this subsection, critical forces $F_{cr}$ are determined within both of the methods. The critical forces for the first buckling mode related to a number of layers are given in Table 4. It was noticed that with an increasing number of layers and a step-variable gradation (according to distributions as in Figure 2), the critical forces grew slightly (0.2–0.3% at most). This trend was also retained in the case of the SAM. In addition, the SAM for five layers showed the greatest discrepancy in the critical forces, which almost disappeared for more layers (even below 0.1%). The critical forces for the clamped plate were only estimated with the FEM. Taking into consideration an increase in the ceramics thickness (changing from 0.2 mm to 1 mm), the critical forces grew as well (Table 5).

This is caused by a significantly higher Young’s modulus and a growing thickness of ceramics. Nevertheless, a higher modulus of ceramics than the modulus of aluminium results in a non-symmetric distribution of the force with regard to the neutral and geometric axis, which can faster yield critical loads. Besides, it should be mentioned that very close results were achieved for both of the methods applied. As expected, the clamped plate can withstand more than twice as high load as in the case of the simply supported plate.

### 3.2. Post-Buckling Behavior of the Plate

In this subsection, the results concern an assessment of the plate behavior in the post-buckling state with reference to different parameters. Figure 4a,b depict a deflection in the center of the plate vs. the static compression load for the SSSS plate (in the first stage, one half-wave was noticed). The curves are plotted for both the methods under consideration. Figure 4a presents an influence of the layer number on the plate stability (from five to 15 layers with a step-variable gradation of material properties according to Figure 2). In the case of a change in the number of layers, any significant difference cannot be seen in the obtained curves. It seems that the overall bending stiffness for the considered cases provides the same effect. In a comparison of the results based on the two methods, a major discrepancy can be noticed. First of all, the FEM results provide a considerable deflection of the plate before the first critical force is reached. In contrast, in the SAM, a deflection appears just in the vicinity only and grows very fast (compared to the FEM curves). Moreover, in the SAM results, a change in the buckling mode was not observed after applying a higher load, which takes place in the FEM analysis in contrast. Furthermore, deflections in the SAM are greater than deflections in the FEM. The same response of the plate was seen for different variants (Figure 4b) by increasing the thickness of alumina and decreasing the thickness of the FGM. Nevertheless, the trends of curves seem to be very logical because with an increase in ceramics, the deflections of the plate slightly drop—this is due to higher resistance. Apart from these aspects, a jump in the buckling mode occurs on the diagrams at higher loads. Minor differences between the results of the two methods were observed for the SCSC plate (clamped unloaded edges—Figure 5a,b) but the plate analyzed with the FEM deflected faster than in the SAM.

The curves obtained with the FEM are almost identical regardless of a number of layers (Figure 5a), or remain close to each other as far as the considered variants (Var_1-Var_5) are concerned. In the case of the FEM analysis, some shifts of the curves are seen due to a growth in the number of layers. This occurs for the force approximately three times greater than the critical force. Taking into account different variants with an increase of ceramics content, a slightly higher resistance of the plate was observed (as in the previous case). However, for all variants, both for the SSSS plate or for the SCSC plate, a change in buckling modes in the plate occurs in the FEM analysis (in contrast, it does not happen in the SAM). The behavior for the plate clamped on all its edges (CCCC plate) based on the FEM alone was investigated (Figure 6a,b). In the first stage of the plate deflection (Figure 6a), the curves coincide with each other but under the load equal to double critical loads, the plots diverge slightly. For this case, a change in the buckling mode from one half-wave to two half-waves was noticed. For all the cases under consideration (Figure 7a,b), the same similarity can be seen. First of all, the deflection begins at a considerably lower force before the critical load is reached. Secondly, as Figure 7b depicts (the static load was referred to the critical load $F_{cr}$ for the given boundary conditions), for the load corresponding to the three-fold critical force, curves showing the middle deflection are almost the same. Afterwards, the further behavior of the plate results perhaps from the type of support available. The maps of plate deformation (displacements in nodes in direction of Z-axis) for three different boundary conditions and for VAR_1 were presented in Table 6. The color “blue” represents maximum values of displacements in the opposite direction to the Z-axis, but the color “red” denotes the maximum value of displacements in the same direction as the Z-axis (See Figure 3a). For the SSSS plate, a natural deflection (in the middle of the plate) was observed. With an increase in the compression force, the deflection was growing along unloaded edges until two half-waves appeared (under the load corresponding to the approximately six-fold critical force). This buckling mode in the plate did not last for a long time because a fast jump of deflection in three half-waves followed rapidly. In the center, the plate changed the direction of the maximal deflection. Afterwards, the buckling mode of three half-waves was continued until the maximal load was reached. In the case of the SCSC plate, more longitudinal deflection from the beginning due to clamped unloaded edges can be seen. The plate behaves very similarly to the SSSS plate. A change in the buckling mode from one half-wave to two half-waves was noted as well (at this point, the static load was equal to triple critical forces). In the case of the CCCC plate, the deflection is concentrated in the middle of the structure under analysis. In general, the deflection changes from the round shape to an elliptical one along the loaded edges. Finally, under a higher load, one half-wave changes violently into three half-waves.

## 4. Summary

An analytical-numerical and numerical analysis of the ceramics-FGM plate subjected to static load was performed. The functionally step-variable plate has a finite number of layers to reflect a real FGM structure. In addition, an influence of thickness of the ceramic wall on the overall stability of the plate was examined. The problem was solved with two methods, namely: the finite element method and the nonlinear asymptotic Koiter’s method. Moreover, the analysis was extended to consider several boundary conditions of the plate. On the basis of the achieved results, it was concluded that:The gradation of layers in the number of plates from five to 15 revealed a slight influence on the plate stability because the obtained curves in both the methods ran almost identically. It indicates that the overall bending stiffness remains on a comparable level, though a non-symmetrical distribution of normal forces with respect to the neutral axis of the plate exists;in a comparison of critical forces based on the two applied methods, a sufficiently good agreement was achieved (a few percent difference, at most). The SAM gave slightly lower values;in the cases under consideration, the growth in ceramics thickness (from 0.2 mm to 1 mm) played an insignificant role in post-buckling paths. Indeed, the differences in curves are visible but they differ only slightly from one another;a higher discrepancy could be seen by comparing the behavior of the plate obtained using the two methods. Firstly, in the FEM analysis, the plate deflects earlier than in the SAM analysis, but after exceeding the critical load, the SAM indicates a larger deflection. In addition, there is a change in the buckling mode during the plate compression. In contrast to the SAM, the FEM reveals a transformation of the defection function from one half-wave to two or three half-waves;when analyzing the curves obtained for three different boundary conditions, a similarity in the plate deflection up to three-fold overloads was noticed if static loads were referred to their critical buckling loads (see Figure 7b).

## Figures and Tables

**Figure 1 materials-12-04090-f001:**
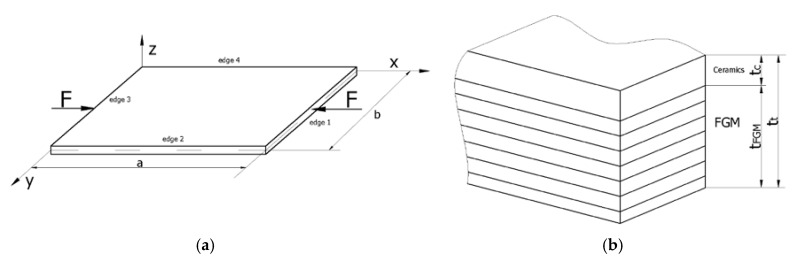
A functionally graded material (FGM) plate with its dimensions and a coordinate system (**a**) and a schematic view of the material distribution (**b**).

**Figure 2 materials-12-04090-f002:**
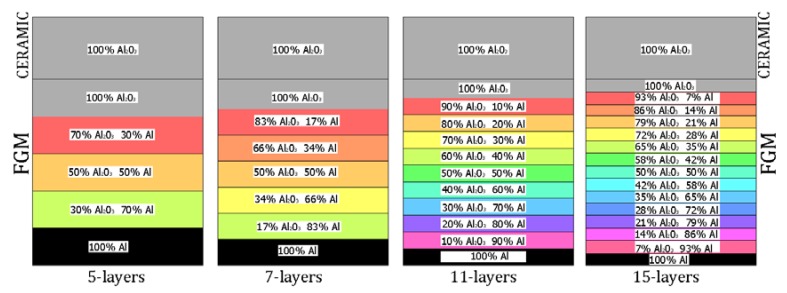
Material distribution versus a number of layers.

**Figure 3 materials-12-04090-f003:**
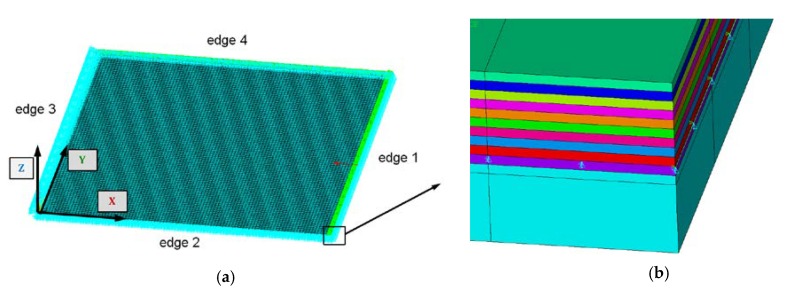
Discrete model with boundary conditions (**a**) and a view of the plate corner (**b**).

**Figure 4 materials-12-04090-f004:**
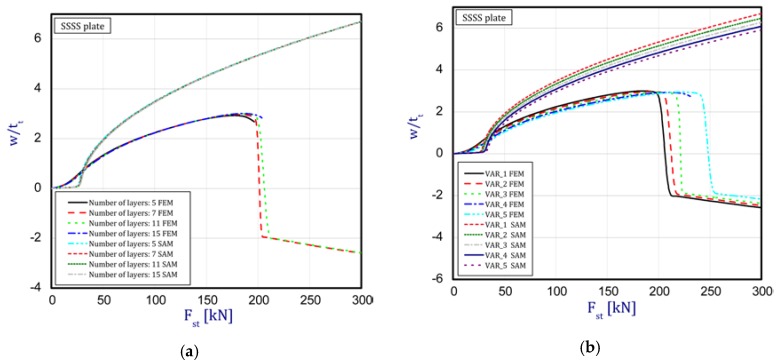
Static load vs. the normalized deflection in the middle of the plate for the plate SSSS: (**a**) a different number of layers (**b**) different variants.

**Figure 5 materials-12-04090-f005:**
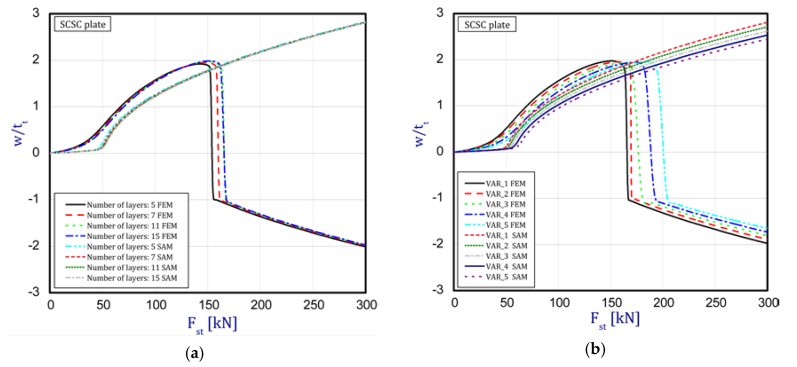
Static load vs. the normalized deflection in the middle of the plate for the plate SCSC: (**a**) a different number of layers (**b**) different variants.

**Figure 6 materials-12-04090-f006:**
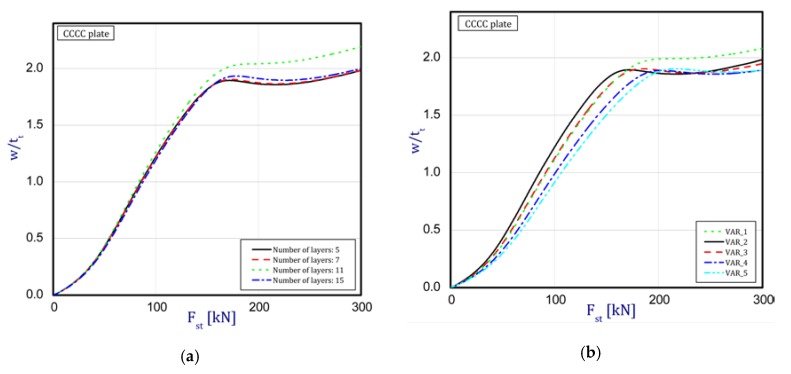
Static load vs. the normalized deflection (FEM) in the middle of the plate for the plate CCCC: (**a**) a different number of layers (**b**) different variants.

**Figure 7 materials-12-04090-f007:**
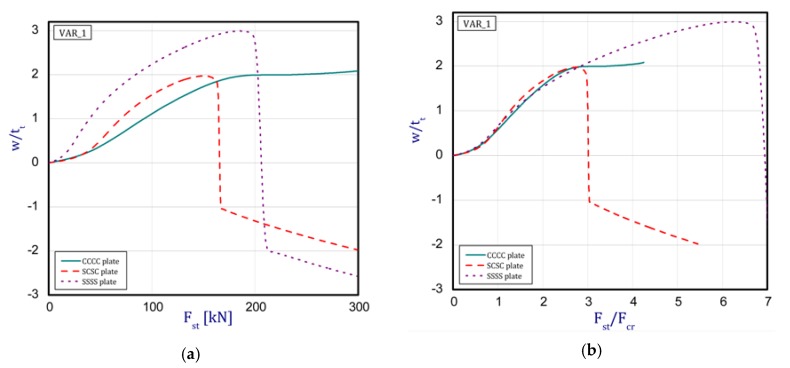
Comparison of the plots (FEM) for Var_1: (**a**) static load vs. the normalized deflection and (**b**) the critical load of static load vs. the normalized deflection.

**Table 1 materials-12-04090-t001:** Thicknesses for individual variants.

Description of Variant	*t_t_* [mm]	*t_c_* [mm]	*t_FGM_* [mm]
Var_1	2	0.2	1.8
Var_2	2	0.4	1.6
Var_3	2	0.6	1.4
Var_4	2	0.8	1.2
Var_5	2	1	1

**Table 2 materials-12-04090-t002:** Material properties of basic constituents.

Components	Young’s Modulus [GPa]	Poisson’s Ratio [-]
Al	70	0.33
Al_2_0_3_	393	0.25

**Table 3 materials-12-04090-t003:** Boundary conditions assumed in the numerical model.

Type of BC	Edge 1	Edge 2	Edge 3	Edge 4
SSSS	u_z_ = 0 u_x_ = moveable applied load Couple degree of freedom for nodes in x-directions	u_y_,u_z_ = 0	u_x_,u_z_ = 0	u_z_ = 0 u_y_ = moveable Couple degree of freedom for nodes in y-directions
SCSC	u_z_ = 0 u_x_ = moveable applied load Couple degree of freedom for nodes in x-directions	u_y_,u_z_ = 0 rot_x_ = 0	u_x_,u_z_ = 0	u_z_ = 0 u_y_ = moveable rot_x_ = 0 Couple degree of freedom for nodes in y-directions
CCCC	u_z_ = 0 u_x_ = moveable rot_y_ = 0 applied load Couple degree of freedom for nodes in x-directions	u_y_,u_z_ = 0 rot_x_ = 0	u_x_,u_z_ = 0 rot_y_ = 0	u_z_ = 0 u_y_ = moveable rot_x_ = 0 Couple degree of freedom for nodes in y-directions

**Table 4 materials-12-04090-t004:** Critical forces versus a number of layers or boundary conditions.

Number of Layers in the FGM (Var_1)	Type of BC	FEM [N]	SAM [N]
5	SSSS	28,994	27,876
7	SSSS	29,179	28,776
11	SSSS	29,663	29,392
15	SSSS	29,830	29,672
5	SCSC	53,491	50,888
7	SCSC	53,895	52,820
11	SCSC	54,970	54,160
15	SCSC	55,339	54,772
5	CCCC	68,584	------
7	CCCC	69,152	------
11	CCCC	70,667	------
15	CCCC	71,186	------

**Table 5 materials-12-04090-t005:** Critical forces versus variants of distributions or boundary conditions.

Variant	Type of BC	FEM [N]	SAM [N]
Var_1	SSSS	29,663	29,392
Var_2	SSSS	30,777	30,524
Var_3	SSSS	31,910	31,716
Var_4	SSSS	33,244	33,140
Var_5	SSSS	34,954	34,956
Var_1	SCSC	54,970	54,160
Var_2	SCSC	57,161	56,384
Var_3	SCSC	59,519	58,836
Var_4	SCSC	62,374	61,840
Var_5	SCSC	66,030	65,688
Var_1	CCCC	70,667	------
Var_2	CCCC	73,571	------
Var_3	CCCC	76,776	------
Var_4	CCCC	80,704	------
Var_5	CCCC	85,734	------

**Table 6 materials-12-04090-t006:** Maps of displacements in direction of Z-axis (Var_1) for the boundary conditions under consideration.

SSSS		SCSC		CCC	
*F_st_ = 16.5 kN*	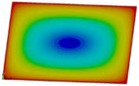	*F_st_ = 16.5 kN*	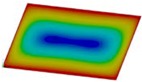	*F_st_ = 16.5 kN*	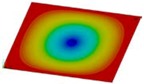
*F_st_ = 88.5 kN*	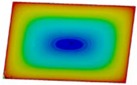	*F_st_ = 88.5 kN*	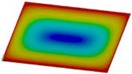	*F_st_ = 88.5 kN*	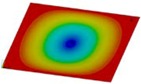
*F_st_ = 154.5 kN*	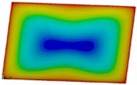	*F_st_ = 154.5 kN*	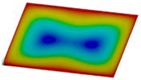	*F_st_ = 154.5 kN*	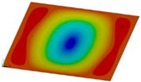
*F_st_ = 190.5 kN*	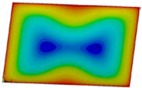	*F_st_ = 183 kN*	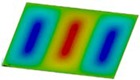	*F_st_ = 190.5 kN*	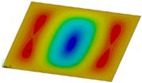
*F_st_ = 300 kN*	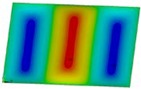	*F_st_ = 300 kN*	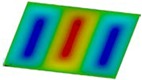	*F_st_ = 300 kN*	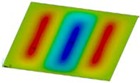

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
