# Peer review of "Analysis of the Functionally Step-Variable Graded Plate Under In-Plane Compression"

_materials, 2019, doi:10.3390/ma12244090_

Round 1

Reviewer 1 Report

In the present work, a comparative analysis of the results of numerical calculations for the mathematical model of FGMs is carried out. The semi-analytical method and the finite element method were chosen as methods for calculating the task. The authors did not develop numerical methods. They used the previously created codes. Based on the calculations, they found coincidences and differences in the results obtained by the two methods. However, I did not see assessments of the quality and benefits of one or another of the methods used. I would recommend the authors supplement the paper with these ratings.

Author Response

Dear Reviewer,

first of all, we’d like to thank for your valuable remarks, which let us improve our paper. Taking into consideration those suggestions, we have done our best to correct our article and to fulfil the requirements of publishable standard. The added or changed text was highlighted on “yellow”.

Reviewer 2 Report

In Czechowski L., Kolakowski Z., "Analysis of the functionally step-variable graded plate under in-plane compression", authors numerically and theoretically investigated the buckling of functionally graded composites. Although the topic is interesting, I believe the manuscript requires significant improvement. It also requires a to be proofread by a professional editor.
Here are my comments:

1) The authors have cited a couple of their previous works. What is the difference between the current work and the authors' previous work, e.g., ref [18]?

2) In the introduction section, lines 29-33, it is required that authors add references to their statements.

3) One of the methods that have been recently used for the fabrication of FGMs is multi-material 3Dprinting. There are several pioneer groups worldwide, such as groups of Prof. Buehler, Prof. Pugno, Prof. Zadpoor, and Prof. Studart. Please refer to their works as well. This is primarily related to your statement on page 2, "Because it seems impossible to achieve an
idealized material distribution (based on the literature survey), ...".

4) You mentioned, "It should be mentioned that actual FGMs differ from one another in contrary to the theoretical description of these materials." Please clarify this statement.

5) In the problem description, page 3, lines 84-87, please rewrite the sentence.

6) What does the number of variants shown in Table 1 express?

7) Please show the coordinate in the model shown in Figure 3.

8) In figure 3, please mention where the zoomed-in plot show?

9) How are the load-displacement curves of these plates different? Why did not you show any of them?

10) Looking at your results (Figures 4, 5, and 7), what is the physical meaning of having negative values for w/tt?

11) What do we understand from Table 6? What do the contours show?

12) How do the different buckling mode shapes can be changed by changing the boundary condition?

Author Response

(The authors gave the same response as above.)

Reviewer 3 Report

The paper is devoted to analysis of the functionally graded multilayered metal-ceramic structure.

Authors have used two methods for study of structure deflection vs static load. I suggest to reject this paper due to the next two main reasons:

There is no explanation in the text from physical point of view why two methods give very different results. There is no explanation the importance of obtained results from practical or fundamental points of view.

Moreover,

There are no enough physical explanations of various boundary conditions used in the paper.

Author Response

(The authors gave the same response as above.)

Round 2

Reviewer 2 Report

The authors have addressed all of my comments. 

Reviewer 3 Report

After author's corrections this manuscript could be published.